# Supramolecular Complexes of Graphene Oxide with Porphyrins: An Interplay between Electronic and Magnetic Properties

**DOI:** 10.3390/molecules24040688

**Published:** 2019-02-14

**Authors:** Kornelia Lewandowska, Natalia Rosiak, Andrzej Bogucki, Judyta Cielecka-Piontek, Mikołaj Mizera, Waldemar Bednarski, Maciej Suchecki, Konrad Szaciłowski

**Affiliations:** 1Institute of Molecular Physics, Polish Academy of Science, ul. Smoluchowskiego 17, 60-179 Poznań, Poland; andrzej.bogucki@ifmpan.poznan.pl (A.B.); waldemar.bednarski@ifmpan.poznan.pl (W.B.); 2Faculty of Technical Physics, Poznan University of Technology, ul. Piotrowo 3, 60-965 Poznań, Poland; natalia3991@wp.pl; 3Department of Pharmacognosy, Faculty of Pharmacy, Poznań University of Medical Sciences, Święcickiego 4, 60-781 Poznań, Poland; mikolajmizera@gmail.com; 4Academic Centre for Materials and Nanotechnology, AGH University of Science and Technology, Al. Mickiewicza 30, 30-059 Kraków, Poland; suchecki@agh.edu.pl

**Keywords:** porphyrins-GO complexes, magnetic material, charge transfer

## Abstract

Graphene oxide (GO) was modified by two modified porphyrins (THPP and TCPP) to form GO–porphyrin hybrids. Spectroscopic measurements demonstrated the formation of stable supramolecular aggregates when mixing two components in solution. The Fourier transform infrared (FTIR) and Raman scattering measurements confirm π-stacking between hydrophobic regions of GO nanoflakes and porphyrin molecules. On the number and the kind of paramagnetic centers generated in pristine GO samples, which originate from spin anomalies at the edges of aromatic domains within GO nanoflakes. More significant changes in electronic properties have been observed in hybrid materials. This is particularly evident in the drastic increase in the number of unpaired electrons for the THPP-GO sample and the decrease in the number of unpaired electrons for the TCPP-GO. The difference of paramagnetic properties of hybrid materials is a consequence of π-stacking between GO and porphyrin rings. An interesting interplay between modifiers and the surface of GO leads to a significant change in electronic structure and magnetic properties of the designed hybrid materials. Based on the selection of molecular counterpart we can affect the behavior of hybrids upon light irradiation in a different manner, which may be useful for the applications in photovoltaics, optoelectronics, and spintronics.

## 1. Introduction

Graphene has many record properties. It is transparent like (or better than) plastic, but conducts heat and electricity better than any metal, it is an elastic film, behaves as an impermeable membrane, and it is chemically inert and stable [1]. Large scale production of graphene is still expensive and difficult, however simple oxidative treatment of pure graphite results of formation of well known, but still somehow mysterious compound—graphite oxide. Under ultrasonic treatment in polar solvents and in the presence of bases it can be exfoliated and dissolved [2]. This new material is usually called graphene oxide (GO). It consists of single nanoflakes of graphene highly decorated with oxygen-containing functional groups. This material, upon deposition on solid substrates shows semiconducting properties (E_g_ ~2.2 eV) with low charge carrier mobility. Graphene oxide is an universal bonding platform well suited for covalent and non-covalent modifications due to the presence of various functional groups (hydroxyl, epoxy, carboxyl, carbonyl) and aromatic domains [3,4,5]. Graphene oxide (GO) and reduced graphene oxide (rGO) based composite with porphyrins are found to be promising materials for light energy conversion, optoelectronics, and photovoltaics because carbon nanomaterials can serve as charge carrier scavengers and molecular-scale conductors, thus facilitating the charge transfer processes involving porphyrin molecules [6,7,8,9,10,11,12,13].

The porphyrins can form the hybrid structure with GO due to various non-covalent interactions such as hydrogen bonding, π–π stacking, hydrophobic interactions, electrostatic interactions, van der Waals forces. etc. Compared with covalent functionalization, noncovalent methods through supramolecular interactions have the advantage of maintaining the unique electrochemical properties of GO and porphyrins as well as simplicity of their fabrication and/or further modification [14]. Recently, noncovalent functionalization of GO has shown exciting potential in terms of loading efficiency and manipulation [15,16]. GO can be easily modified with various redox-active compounds [17] and other carbon nanostructures, including carbon nanotubes [18]. The interest in noncovalent functionalization of GO with porphyrins in particular results from the fact that porphyrins form a planar 18-pi electron ring or macrocycle, have extremely high molar absorption coefficients and show numerous potentially useful photochemical and photophysical properties, including photoinduced energy and electron transfer processes [19,20]. More importantly, it is possible to endow optoelectronic properties of GO when it is combined with planar aromatic molecules such as porphyrins and phthalocyanines [21,22].

The noncovalent stacking of aromatic organic molecules on graphene oxide through π–π interaction is emerging as a promising route to tailor the electronic properties of graphene oxide. In graphene oxide- core-modified-porphyrins and expanded porphyrin hybrids are observed an enhanced nonlinear optical response.

Therefore, in this paper we investigate interactions between graphene oxide and two free-metal porphyrins using various spectroscopic techniques supplemented with quantum-chemical modeling. The basic understanding on the electronic interaction of meso-substituted porphyrin molecules with graphene oxide is very important to design promising devices. In the present work we provide a simple route of preparation of a GO–porphyrin assemblies. A hydrophobic porphyrin derivative was combined with GO on the basis of affinity of p-electron systems [23,24]. The formation of GO-porphyrin composite has been well characterized by ultraviolet–visible spectroscopy (UV-Vis), Fourier transform infrared (FTIR) spectroscopy, Raman scattering and electron paramagnetic resonance (EPR) measurements. The main aim of the study is to understand the nature of interaction between the components as well as elucidate the electronic structure of these new supramolecular entities and the nature of the electron/energy transfer processes between constituents within supramolecular assemblies.

## 2. Results and Discussion

### 2.1. Optical Spectroscopy

The absorption spectra of the two porphyrins THPP and TCPP show a typical Soret band and four Q bands (Figure 1a). The strong band at 422 nm for THPP and at 418 nm for TCPP was assigned to the Soret band arising from the transition to the second excited state (S0 → S2) and the other four absorption maxima at 518, 556, 595, and 652 nm for THPP; 514, 549, 589, and 645 nm for TCPP were attributed to the Q bands corresponding to weak transition to the first excited state (S0 → S1). The Soret and the Q bands both arise from π–π* transitions and can be explained by considering the four frontier orbitals (HOMO and LUMO orbitals) [25,26].

The UV-Vis absorption spectra of GO are characterized by two peaks at 232 nm and 295 nm. The first peak comes from the π–π* transition and is related to the C=C bond in an aromatic ring and the second one is due to the *n*–π* transition of the C=O bond [27,28,29].

The fluorescence spectra of the metal-free porphyrins at a concentration of about 10^−6^ M are shown in Figure 1b. It can be seen that studied porphyrins display two emission peaks at 663 and 726 nm (THPP); 653 and 716 nm (TCPP), respectively. This spectral pattern is characteristic for the monomeric porphyrin molecules. Comparing the fluorescence spectra of THPP and TCPP one can observe, that the luminescence intensity of THPP is much stronger than that of TCPP. This is caused by hydroxyl and carboxyl groups in the two porphyrins, which influence the increase and decrease in the average electron density of the conjugated porphyrin system, respectively. It causes the facilitation of the intersystem crossing (ISC) of S1→T1, therefore the fluorescence intensity of TCPP is weaker compared to the case of THPP [30].

Figure 2a shows the UV-vis spectra of the porphyrins-GO complex in the presence of different concentrations of GO. With a gradual increase of GO in the hybrid systems, absorption peak positions undergo small changes of about 1–2 nm. This observation suggests that the porphyrin geometry is not altered after the composite formation with GO, especially that the supramolecular interaction does not induce any significant structural deformations of porphyrin molecules. It also indicated that the stacking attachment of the GO moiety has not perturbed the ground electronic state of the porphyrin and is strong enough to make the material stable also in the solid state. These results corroborate not only the non-covalent linkage of porphyrin with the GO sheets but also electronic interaction between the two species in the ground state [31]. The alternative aggregation potential induced by the H bond is excluded because no significant shifts in the absorption bands have been observed and this is further substantiated by IR and Raman spectroscopies. These results indicate that electrostatic interaction has a weak effect on the supramolecular assembly of porphyrins molecules on GO [32]. These results are in agreement with studies based on other hybrid systems consisting of porphyrins covalently grafted to carbon nanotubes and nanohorns [33,34,35,36,37,38,39].

To further demonstrate that porphyrins could be assembled onto the GO, we used fluorescence spectroscopy to investigate the GO-porphyrin systems. Figure 3 shows the fluorescence spectra of two metal-free porphyrin-GO complexes formed in various concentrations of GO at the excitation wavelength of 418 nm. THPP, TCPP have a broad emission comprising of two unresolved Q(0,0) and Q(0,1) bands at ca. 663 and 726 nm (THPP); 653 and 716 nm (TCPP), respectively. Vergeldt et al. postulated that coalescence of the Q-bands is caused by the mixing of the first excited-state S1 of the porphyrin and the charge transfer (CT) state in which an electron is transferred from the porphyrin macrocycle to the pyridinium substituent of the porphyrin. For coplanar orientation of the pyridinium groups and the porphyrin macrocycle, electronic coupling between the S1 and the CT states increases [40]. For our material a similar case takes place and the electron must be transferred from the porphyrin macrocycle to the peripheral aryl groups.

For the porphyrins-GO systems the emission peak positions remain unchanged, but the photoluminescence intensity decreases with increasing concentrations of GO, which indicates the existence of free and adsorbed porphyrins in equilibrium. The plot of *F*_0_/*F* to *Q* (quencher concentration) is neither linear nor parabolic. Accordingly, the static/dynamic quenching coexisted and related constant were determined from the following Stern–Volmer equation [41]:(1)$$\frac{F_{0}}{F}=1+\left(K_{d}{+K}_{S}\right)\left[Q\right]{+K}_{d}\cdot K_{S}{\left[Q\right]}^{2}$$
where *F*_0_ and *F* denote the fluorescence intensities of the fluorescent substance in the absence and in the presence of quencher concentration of [*Q*]. *K*_d_ and *K*_s_ are the dynamic quenching constant and static quenching constant, respectively.

The results of the calculation of the THPP-GO was *K*_s_ = 196.76, *K*_d_ = 0.09 and of the TCPP-GO *K*_s_ = 31.61 and *K*_d_ = 0.02, respectively. The *K*_s_ is much bigger than *K*_d_. This indicates that non-fluorescent complex formation was the main reason for decreasing fluorescence intensity [42]. The quenching efficiency was calculated as 89 and 51%, for THPP-GO, and TCPP-GO complexes, respectively. This indicates that the THPP derivative interacts stronger with GO sheets. The fluorescence quenching may be attributed to the photoinduced electron transfer between porphyrin molecules and GO. The Förster energy transfer should be ruled out because there is no significant overlap between the emission spectrum of porphyrin and absorption spectrum of GO. The Dexter mechanism is also not very probable due to the large distance between the counterparts and the lack of any covalent bonds that would provide a platform for electron exchange. This hypothesis is in agreement with the electron transfer from porphyrins to GO through the π–π interaction observed in the cases of nanotubes or fullerenes [37,43,44,45,46,47,48,49]. These results suggest that the singlet excited state of porphyrin interacts with GO resulting in a weak or non-emissive complex. Strong interactions between GO and the excited state of porphyrin, as well as large specific interfacial area in the two dimensional plane of GO might be responsible for attenuated fluorescence emission intensity in the complex via the formation of multiple hetero-junctions between porphyrin and graphene oxide. This efficient quenching of fluorescence emission indicates that the porphyrin-GO complex can be used as an active material for optoelectronic applications working on the basis of photoinduced electron transfer processes. Furthermore, the calculated electronic structure (*vide infra*) supports this hypothesis.

The electron density of the porphyrins is shown in Figure 4 where the optimized geometries and the HOMO and LUMO surfaces of the studied porphyrins are presented. For the two porphyrins the HOMO orbital is delocalized over the methine bridges and nitrogen atoms. This means that these atoms are the most likely to donate electrons. The main contributing factor to the HOMO is the porphyrin ring because the substituent groups (in all compounds) appear to have negligible HOMO density. The LUMO orbital is delocalized over the methine bridges and pyrrole rings. DFT calculations indicate that the lower-energy band corresponds to the HOMO→LUMO transition, whereas the band at about 420 nm corresponds to the HOMO→LUMO + 1 transition. This data led to an estimate of the HOMO-LUMO gap of 1.9 and 1.92 eV for THPP and TCPP, respectively. According to DFT calculations, the difference between the HOMO and LUMO energies is 2.66 and 2.70 eV, which is in perfect agreement with the electrochemical data. The electrochemically evaluated HOMO-LUMO gap (the difference between the oxidation and reduction potentials) amounted to 1.6 and 1.7 eV. The discrepancy between the spectroscopic and the calculated band-gap originates from the electron–hole pair binding energy [50] and large reorganization energy [51].

Other quantum chemical parameters were computed to have more insights into the reactivity and selectivity of the porphyrins. The molecular orbital energies (i.e., E_HOMO_ and E_LUMO_) can provide information about the reactivity of chemical species and are often associated with the electron donating ability of a molecule [52,53,54]. The higher E_HOMO_ value indicates higher tendency of the molecule to donate electron (s) to the appropriate acceptor species with low energy and empty/partially filled atomic/molecular orbitals. The results obtained by DFT calculations and experimental measurements show that THPP has the higher value of E_HOMO_ than TCPP (see Figure 4). This effect is caused by different substituents at terminal aryl groups. It is well established, that in such systems the -OH group is an electron donor and -COOH group is an electron acceptor. Therefore, the THPP derivative has a higher tendency to donate electrons than TCPP, which may have significant electron acceptor character.

The energy levels of the GO structure were calculated and the corresponding electronic orbitals are presented in Figure 4. The calculated HOMO-LUMO band gap of GO is about 0.61 eV. This value corresponds to absorption bands in NIR/MIR IR range, and strongly heterogeneous atomic and electronic structures of GO indicate that fluorescence of GO arises from recombination of electron–hole pairs in localized electronic states originating from various possible configurations, rather than from band-edge transitions as is the case of typical semiconductors.

Due to GO cluster size the values of the energy gap are quite small. As published by Lonfat et al. the HOMO-LUMO energy gap decreases with the increase of the system size [55]. Molecular orbitals (HOMO and LUMO) and electron density are very useful for predicting the most reactive position in π–electron systems and also explain several types of reaction in conjugated systems. The conjugated molecules are characterized by small HOMO, LUMO seperation, which is the result of a significant degree of charge transfer from end-capping electron-donor groups to the efficient electron acceptor groups through π–conjugated path. In the THPP-GO and TCPP-GO complexes the energy gap is 0.33 eV and 0.37 eV, respectively, and is eight times smaller than energy gap of free-metal porphyrins and smaller than the GO as well. The absence of any substantial red-shifted absorptions spectrum for porphyrin in the porphyrin-GO complex indicates that its HOMO-LUMO transition is essentially unaffected by the formation of supramolecular hybrids with graphene oxide, which is also confirmed by DFT calculations [56].

### 2.2. Vibrational Spectroscopy

In order to interpret the experimental results of IR absorption and Raman scattering investigations into the quantum chemical calculations were performed and the DFT-level calculations of normal mode frequencies and intensities were performed. In Figure 5 the experimental and calculated infrared (Figure 5a,b) and Raman (Figure 5c,d) spectra of THPP, GO and THPP-GO and TCPP, GO, and TCPP-GO are presented respectively. The most important bands are also collated in Table 1 and Table 2. In general, conformity between the calculated and experimental spectra is quite good. The wavenumber shifts between them are typical and arise from approximations used in the computational procedure. Mainly the anharmonicity of vibrations and environment of the molecules which are neglected in our calculations cause those shifts. Panels in Figure 5 show only selected spectral ranges where the differences between the spectra of both molecules are presented.

For the THPP in IR absorption spectrum at low frequencies the bands are located at 535, 560, 597, 729, 804, and 983 cm^−1^ (Table 1). The first three bands are related to the wagging vibration of the C-H bonds at the benzene rings and deformation of the porphyrin ring. The next three bands are observed for both porphyrins and are associated with wagging vibration of the N-H bonds. They have also additional components related to the breathing of the pyrrole ring in the porphyrin ring. In the IR absorption spectra there are observed quite strong bands related to the stretching vibration of the C-C and C=C bonds in the porphyrin and benzene rings. They are located at 1465, 1508, 1586, and 1605 cm^−1^ for THPP, and at 1473, 1505, 1564, and 1605 cm^−1^ for TCPP (Table 2). For both samples the bands at about 1230 cm^−1^ corresponding to the stretching vibration of the C-C bond between the porphyrin ring and aryl groups are also visible. This band is also good visible in the Raman scattering spectra and is located at 1234 cm^−1^ for THPP, and at 1231 cm^−1^ for TCPP. At the 1381/1358 cm^−1^, 1459/1440 cm^−1^, 1516/1495 cm^−1^ and 1544/1545 cm^−1^, and 1608/1605 cm^−1^ observed for THPP/TCPP, respectively are located the bands related mainly to the stretching vibration of the C-C, C=C and C-N bonds, rocking vibration of the C-H and N-H bonds (see Table 1 and Table 2). In the range of 200–400 cm^−1^ the Raman scattering spectra of both porphyrins are similar. For example at this frequencies the bands are observed at 334/319 and 417/410 cm^−1^ related to breathing of the porphyrin ring and deformation of the benzene rings, respectively. We cannot forget that the nature of the spectrum in this range are also affected by lattice vibrations.

Two different aryl groups linked to the porphyrin have an influence on IR and Raman spectra. In the IR absorption spectra for THPP there are observed quite strong bands at 1169 cm^−1^ and 1263 cm^−1^ associated with the bending vibration of the C-O-H and stretching vibration of the C-O bonds. In Raman scattering spectra at 1171 cm^−1^ the band related to the bending vibration of the C-O-C bonds in the hydroxyl group is located. For TCPP the very strong band at 1691 cm^−1^ corresponds to the stretching vibration of the C=O bond in the carboxyl group. Besides that some of the bands have additional components, for example the bands at 1176 and 1220 cm^−1^ are related to the bending vibration of the C-O-H bonds too. This and the other vibration in the aryl groups have influence on position and shape of bands related to the vibration in the porphyrin ring. The bands related to stretching vibration of C-H and O-H bonds located above 3000 cm^−1^ are less intense than the interaction between porphyrin molecules suggests.

FTIR spectroscopy is recognized as an important tool for the characterization of functional groups and in the case of GO has supported the presence of hydroxyl groups. The bands at: 1056, 1223, 1380, 1616, and 1727 cm^−1^ are related to the stretching vibration of the C-O, C-O-C, C-OH, and C=O, respectively. Broad peak at 3000–3500 cm^−1^ is corresponding to the stretching vibration of the C-H and O-H [15,57].

The Raman spectrum of GO displays a D band at ∼1334 cm^−1^ and a broad G-band at ∼1603 cm^−1^. Thus, the integrated intensity ratio of the D- and G-bands (ID/IG) indicates the oxidation degree and the size of sp^2^ ring clusters in a network of sp^3^ and sp^2^ bonded carbon. When porphyrin was assembled with GO, ID/IG decreased from 1.16 to 0.96 and 0.94 for THPP-GO and TCPP-GO, respectively.This fact indicates that the extended π–electron structure on the surface of GO is formed on interactions with porphyrin molecules. Therefore it can be concluded that porphyrin molecules are assembled on GO through π–π stacking interactions [58,59].

The IR absorption spectra of the THPP-GO and TCPP-GO (Figure 5) exhibit characteristic bands for the porphyrins, but the maxima of the bands are shifted (see Table 1 and Table 2) and their shape and relative intensity are different. The biggest changes are observed for the range between 1050–1700 cm^−1^. Within this range one can observe mostly bands related to: deformation vibration of the C-H and N-H bonds in the porphyrins rings, bending vibration, stretching vibration of the C-C, C-N, and C=C C=O bonds in porphyrins and finally the bending vibration of the C-O-H bonds in aryl groups in both porphyrins. The big shifts are observed to the bands related to the stretching vibration of the C-C and C=C bonds in porphyrins rings. For THPP-GO these bands are located at 1442, 1477, and 1511 cm^−1^, and for TCPP-GO at 1418, 1470, and 1567 cm^−1^. On the lower frequencies bands associated to the deformation of the C-H and N-H bonds are located and changes in the band maximum position and intensity of the bands are visible. These changes may stem from the stiffening porphyrin molecule located above the plane of the graphene oxide. The most characteristic changes for the THPP are visible at the band 1223 cm^−1^ that is shifted in the THPP-GO hybrid to the 1233 cm^−1^ and is related to the bending vibration of the C-O-H bond. The other bands that have additional components associated with the vibration of the C-O-H bond are also shifted, for example 1346, and 1433 cm^−1^. The same changes are observed for the TCPP-GO. In this case the bands related to this vibration are located at 1176, 1221, and 1270 cm^−1^ for TCPP and at 1181, 1225, and 1279 cm^−1^ for TCPP-GO, respectively. For THPP-GO hybrid the bands related to the stretching vibration of the C-O bond in the aryl group is shifted about 18 cm^−1^ in comparison with THPP and is located at 1281 cm^−1^. For THPP-GO we also observe a new and quite strong band at 1377 cm^−1^, which is probably related to the deformation vibration of the C-H bonds.

In the hybrid structures we also observe the changes in the characteristic bands for GO. The band associated with the stretching vibration of the C-O and C-O-C bonds located at 1056, 1380, and 1616 cm^−1^ in the hybrid almost disappeared. This suggests the formation of a covalent bond between the porphyrin and the graphene oxide via carboxyl and hydroxyl groups [58]. On the other hand, the bands related to the stretching vibration of the C=O bonds at 1727 cm^−1^ are shifted to 1714 cm^−1^ and 1720 cm^−1^ when THPP and TCPP assemble with GO.

In the range above 3000 cm^−1^ there are located the bands related to the stretching vibration of the C-H and O-H bonds one can observe also very significant changes. In hybrid structures the bands are more exposed and separated. The band associated with O-H vibration is strongly shifted even about 58 cm^−1^ in THPP-GO and is observed at 3312 and 3379 cm^−1^ in THPP-GO and TCPP-GO, respectively. In the spectrum of THPP-GO and TCPP-GO, the peak at 2924/2916 cm-1 spons to the sp3 C-H characteristic stretching band. These results clearly indicate that the THPP and TCPP molecules have been non-covalently bonded to the graphene oxide by π–π stacking and are not engaged in any other (e.g., hydrogen bonding) interaction.

As mentioned for GO two strong bands at 1334 (D) and 1603 cm^−1^ (G) [59] are observed. For the THPP-GO in the range of 1300–1600 cm^−1^ there are observed broader ranges corresponding to the pure GO. For the spectrum of THPP-GO are observed almost all characteristic bands for THPP too. The intensity of bands at 417 and 814 cm^−1^ decreases. This band corresponds to the deformation of the benzene ring and the bending vibration of the C-C-C bonds in porphyrins rings and wagging vibration of N-H bonds, and the small stretching vibration of the C-O bond in the aryl group. Other bands are shifted about 1–22 cm^−1^. The Raman scattering spectrum of the TCPP-GO hybrid system does not contain many distinct bands coming from pure porphyrin and is dominated by the two bands of graphene oxide. While the position of the G-band is the same, the D-band is shifted to 1363 cm^−1^ for the TCPP-GO complex. The same bands derived from pure porphyrins have a low intensity. For example the bands related to the breathing of the porphyrin ring of TCPP are located at 319 and 999 cm^−1^ and in the TCPP-GO they are shifted to 324 and 1003 cm^−1^. The band corresponding to the wagging vibration of the N-H bond of TCPP is at 671 cm^−1^ and of TCPP-GO is shifted to position 690 cm^−1^. The strong band registered to porphyrin at 1231 cm^−1^ and corresponding to the stretching vibration of the C-C bond between the porphyrin ring and aryl substituent is shifted to 1236 cm^−1^ and almost invisible. The similar changes are also observed for the band at 1545 cm^−1^ associated with the stretching vibration of C = C bonds in the porphyrin ring. These results obtained from FTIR indicate the bonds formation between the porphyrins and GO structures. These hybrids are formed by the hydroxyl and carboxyl groups.

On the other hand, when porphyrin was assembled with GO, ID/IG decreased from 1.16 to 0.96, 0.94 for THPP-GO and TCPP-GO, respectively. This indicates that the big π-electron structure on the surface of GO was formed. Thus, porphyrins could be assembled on GO through π–π stacking interaction [60,61]. The D/G intensity ratio of hybrids decreases, indicating that the graphene carbons of the composites contain fewer defects than the GO. The decreasing ID/IG ratio corresponding to the presence of GO domains sp^2^ hybridized too. Lomeda et al. [62] reported that the decreased ID/IG ratio is a result of the functionalization of GO. In addition, the hybrid shows a shift of the D peak towards higher wave-numbers, which confirms the ordered porphyrin-GO assembly.

The changes in vibrational spectra suggested some possible electron transfer processes, which potentially could lead to the reduction of graphene oxide with the THPP. In order to verify this hypothesis a set of EPR spectra of GO and GO complexes with two studied porphyrins have been recorded within a large temperature range (20–300 K).

### 2.3. Magnetism

As it was shown in ultrasonic irradiation and some reagents can create additional hydroxyl radicals in solution, which should affect the magnetic properties of the final material. Therefore, using EPR we studied both GO obtained from water or water/THF mixtures. Figure 6 presents EPR spectra of powdered GO samples obtained after evaporation of sonificated GO from H_2_O or H_2_O & THF solutions. At 300 K the spectra consist of an almost symmetrical, strong Lorentzian line (linewidth ΔB_pp_ = 1.30 ± 0.05 Gs) with g-factor 2.0029 ± 0.0002, similarly as previously observed [63,64]. The difference in line intensity for both samples (see the gain of the amplifier on Figure 6) indicates that THF used in solution has some impact on the number and/or type of paramagnetic centers in GO, which may be associated with different unfolding/deaggregation in different solvent systems. For both samples, especially at low temperature, one can observe weak satellite lines resulting from electron-proton interaction [65]. Temperature study of spin susceptibility χ_EPR_ were performed to obtain information about the type of paramagnetic centers in the samples as presented in Figure 7.

Experimental points of spin susceptibility (proportional to double integrated EPR signal), denoted in Figure 7 as squares, can by approximated by the following equation:(2)$$\chi_{EPR}(T)=A+\frac{C_{1}}{T}+\frac{C_{2}}{T}e^{\frac{-\Delta E}{k_{B}T}}$$
where *A* is the constant, *T* is the temperature, *C*_1_ and *C*_2_ are the Curie constants, *k*_B_ is the Boltzmann constant and ΔE is the activation energy.

The first term in Equation (2) describes delocalized electrons (Pauli susceptibility), and the second and third ones describe localized paramagnetic centers for Curie and thermal activated spin susceptibility, respectively. The last part of spin susceptibility may result from antiferromagnetic coupled neighboring defects at low temperatures. Table 3 presents the parameters obtained from fit of Equation (2) to experimental data. Over the whole temperature range the total spin susceptibility is higher for the sample prepared from H_2_O&THF solution. This is due to the strong oxidation processes that manifest as a rapid growth of the Curie component and decreasing of Pauli and activated spin susceptibility.

Figure 8 presents the EPR spectra of hybrid systems porphyrin-GO with g-factors 2.0028 ± 0.0002 and 2.0029 ± 0.0002 at 300 K for THPP-GO and TCPP-GO, respectively. The linewidths of porphyrin-GO are only slightly broadened in comparison to GO: ΔB_pp_ = 1.45 ± 0.05 for THPP-GO and ΔB_pp_ = 1.42 ± 0.05 Gs for TCPP-GO. The spectra of hybrid samples are not a superposition of GO and porphyrins indicate strong interactions between the components.

The mutual influence of GO and porphyrins on electronic properties can be tracked by comparison of spin susceptibility versus temperature. As shown in Figure 9, Equation (2) with parameters presented in Table 3 describes satisfactorily all experimental data. Spin susceptibility of hybrid samples, similar as in the case of EPR spectra, is not a superposition of GO and porphyrins. For TCPP-GO hybrid sample, there is a noticeable decrease of susceptibility in comparison with pure GO from H_2_O&THF. In our case this reduction does not exceed experimental error if one compares Pauli part of the spin concentration of TCPP-GO (0.37 ± 0.02 × 10^17^ electrons/g) with GO from H_2_O&THF (0.45 ± 0.17 × 10^17^ electrons/g).

A different behaviour of spin susceptibility to those described for TCPP-GO shows a THPP-GO hybrid. One can observe a significant increase of spin susceptibility (Figure 9) in the whole temperature range in comparison with GO from H_2_O&THF. This phenomenon is especially visible for Pauli part for which the spin concentration (4.58 ± 0.24 × 10^17^ electrons/g) exceeds one order of magnitude in comparison with GO from H_2_O&THF (see Table 3). Such an increase in the number of delocalized electrons of THPP-GO indicates on expanding of π structure due to the assembling of THPP on GO through π–π stacking interaction.

This striking magnetic behaviour of graphene oxide and its supramolecular assemblies with porphyrins can be explained on the basis of DFT calculations under open shell conditions. It is a well-known fact that higher acenes as well as other large aromatic conjugated systems show significant radical character despite even number of valence electrons. Due to magnetic effects large aromatic systems become di- or even polyradicals, however in total the bear singlet configuration and the electrons are usually antiferromagnetically coupled [66,67]. This effect is, among others, responsible for very low stability of pentacene and higher acenes, graphene folding and other phenomena, but also gives rise to prospective applications of carbon-based materials in spintronics [68]. In the case of graphene oxide, however, some reports predict ferromagnetic coupling of electrons belonging to the edges of aromatic domains. Various carbon nanomaterials are expected to be magnetic [69]. Graphene oxide, especially containing epoxy groups, is also expected to be magnetic with significant ferromagnetic coupling [70,71,72,73]. Furthermore, these magnetic properties are much more evident in the case of reduced graphene oxide [74,75] and the measurements of magnetic properties of graphene oxide were reported as a useful tool for monitoring the reduction process [73,76].

DFT calculations of studied systems indicate significant spin impurities within pristine graphene oxide (Figure 10b), its TCPP complex (Figure 10a), and the THPP-GO complex (Figure 10c).

These results are fully consistent with observed magnetic properties. In clearly indicated the role of electron donor and electron acceptor substituents in porphyrin rings on magnetic properties of graphene oxide. It is evident that even a subtle change results in a very significant change. Moreover, stronger π electron interaction (as in the case of THPP) results in the partial delocalization of unpaired electrons over the porphyrin ring as shown in Figure 10c. These theoretical results do not reproduce the experimental data quantitatively, but support qualitatively the observed trends. It is caused by the low level of applied theory (due to the size of the molecule other methods are demand much higher computing power). Therefore one can conclude that observed magnetic behavior, i.e., increased spin susceptibility in the case of THPP and decreased in the case of TCPP are fully consistent with simple computational model.

## 3. Materials and Methods

### 3.1. Materials

The 5,10,15,20-Tetrakis (4-hydroxyphenyl)-21*H*,23*H*-porphine (THPP), and 5,10,15,20-Tetrakis(4-carboxyphenyl) 21*H*,23*H*-porphine (TCPP) were purchased from Sigma-Aldrich (Poznań, Poland). Their molecular structures are presented in Figure 11.

Graphene oxide was synthesized from natural graphite powder by the modified Hummers method. Graphite powder (3 g) and KNO_3_ (3 g) were added to concentrated H_2_SO_4_ (90 mL) and the mixture was stirred in ice bath. Small portions of the oxidizing agent KMnO_4_ (9 g) were added slowly in order to keep the suspension temperature below 2 °C. The reaction mixture was maintained at approx. 0 °C and it was vigorously stirred for 15 min resulting in the increase of temperature to 35 °C. The stirring was continued for 7 h. Afterwards, 90 mL of distilled water was slowly added and the mixture temperature increased to 80 °C. After 15 min the suspension was cooled to room temperature and the reaction was quenched by the addition of 12 mL of H_2_O_2_ (30%). In the first step of the purification process the diluted (with 250 mL of distilled water) supernatant was poured off. The precipitant was rinsed two times with 250 mL of distilled water and the supernatant was removed. Then the product was centrifuged in 1 mol/dm^3^ HCl aqueous solution at 6000 rpm for 3 min three times (to remove manganese compounds) followed by decantation. Afterwards, the process was repeated four times with distilled water. The final product was dried at room temperature.

For THPP-GO, TCPP-GO composites, we have synthesized distilled water (H_2_O) soluble graphene oxide. Briefly, 4 mg of as prepared graphene oxide (GO) was dissolved in 20 mL H_2_O under mild ultrasonication until brown solution appeared.

The solutions of porphyrins modifiers were prepared by the dissolution of porphyrin in 50 mL THF to achieve a concentration of 10^−5^ M. In the next step 0.2 mL of the porphyrin solution were mixed with GO distilled water solution with increasing concentrations (from 0 to 1.64 μg/mL, then sonicated for 15, 30, and 60 min, thus yielding final products labelled THPP-GO and TCPP-GO, respectively.

### 3.2. Instrumentation

Diffuse reflectance spectra were measured on Lambda 750 (Perkin Elmer, Norwalk, CT, USA) spectrophotometer, sapmles were dispersed in spectrally pure barium sulfate prior to measurements, the same materials were used as reference samples. For the infrared spectroscopy, Raman scattering, and EPR spectroscopy measurements the complexes were prepared by the dissolution of porphyrin in 2.5 mL THF (10^−3^ M). In the next step 0.25 or 0.75 mL of the porphyrins solution were mixed (sonification) with 2.5 mL GO dissolved in H_2_O. After evaporation of the solution, a hybrid material was obtained in the form of a powder. The infrared absorption spectra were recorded using an FTIR Bruker Equinox 55 spectrometer (Bruker Optics, Ettlingen, Germany) equipped with Bruker Hyperion 1000 microscope in the range 400–4500 cm^−1^ at room temperature. Raman scattering spectra of the investigated compounds were recorded using LabRAM HR 800 spectrophotometer (HORIBA Jobin Yvon, Montpellier, France) with excitation λ_exc_ = 458 nm. The power of the laser beam at the sample in all cases was less than 1 mW with a power density of about 3 × 10^8^ mW cm^−2^. Such low power density was necessary to avoid decomposition of the sample. The infrared absorption and Raman scattering was measured from pure samples not dispersed in the matrix. EPR studies were performed using BRUKER ELEXSYS E500 spectrometer (Bruker, Billerica, MA, USA) in the temperature range 20–300 K. EPR spectrometer was equipped with Super High Sensitivity Probe head and ER 036TM NMR-Teslameter. Sample temperature was controlled in OXFORD ESR900 cryostat by ITC503S controller (Tubney Woods, Abingdon Oxfordshire, UK). Low microwave power (0.6 mW) was applied to avoid saturation effect.

### 3.3. Computation

In order to realize an optimal structure of the compounds and to interpret the experimental results of UV-Vis, IR absorption and Raman scattering investigations quantum chemical calculations were performed. The molecular geometries were optimized using the Density Functional Theory (DFT) method with B3LYP hybrid functional and 3–21G basis set. Due to large atom numbers and open shell calculations larger basis sets could not be used due to the limitations of our cluster. The calculations of normal mode frequencies and intensities were also performed. All calculations were made using a Gaussian 09 package (Wallingford, CT USA) [77]. The GaussView (Wallingford, CT USA, Version E01) [78] program was used to propose an initial geometry of investigated molecules and for visual inspection of the normal modes.

## 4. Conclusions

We successfully developed a simple and effective method for preparing porphyrin-GO hybrids that have donor-acceptor properties and are characterized by the charge transfer between porphyrin and graphene oxide. Various spectroscopic studies indicate significant non-covalent interaction, which does not modify optical properties of porphyrin itself (except of significant photoluminescence quenching) but dramatically changes magnetic properties. This effect is attributed to subtle tuning of graphene domain magnetism induced by electron density modulation by electron donor or electron acceptor substituents.

Fluorescence quenching with increasing GO content shows electron transfer from the porphyrin molecule to GO, and an additional band at 627 nm indicates the interaction between porphyrin and graphene oxide. The creation of a hybrid system can also be demonstrated by recorded IR absorption and Raman scattering spectra. The appearance of an additional band for example with a maximum at 1731 cm^−1^ associated with the oscillation of C=O bonds and changes in the spectrum above 3000 cm^−1^ suggest the interactions between porphyrin and GO through the carboxyl and hydroxyl groups. We demonstrated the expanding of π structure due to the assembling of porphyrin on GO through a π–π stacking interaction too. The EPR spectroscopy showed that porphyrins have a significant influence on the electron properties of hybrid materials. This is particularly manifested for THPP in the part of Pauli’s magnetic susceptibility describing delocalized electrons, which exerts a profound effect of the magnetic properties of the hybrid.

We believe that our facile method provides a simple and practical means of functionalization of graphene oxide and thus leads to the further development of a new class of GO based structures, finding various applications, especially in spintronics and optoelectronics.

## Figures and Tables

**Figure 1 molecules-24-00688-f001:**
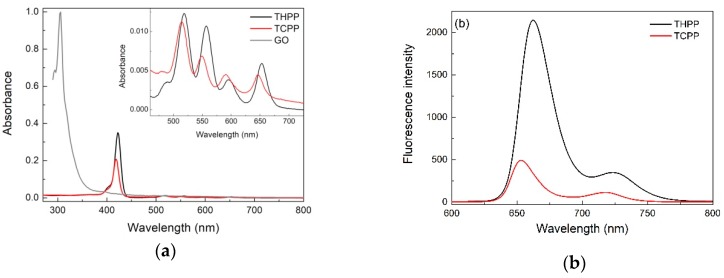
Absorption (**a**) and fluorescence (**b**) spectra of two porphyrins THPP (black) and TCPP (red).

**Figure 2 molecules-24-00688-f002:**
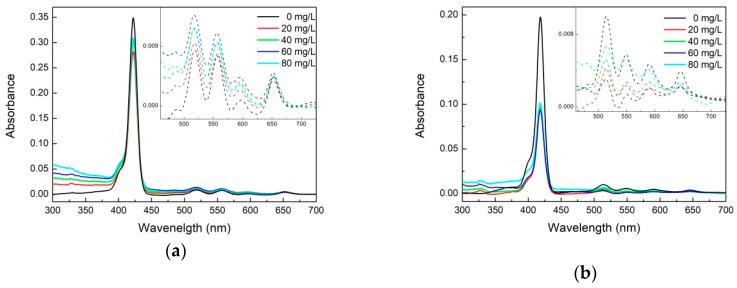
Ultraviolet–visible spectroscopy (UV-Vis) spectra of metal-free porphyrins THPP (**a**), TCPP (**b**), 10^−4^ M with graphene oxide (GO) in different concentrations of GO red—20 mg/L, green—40 mg/L, blue—60 mg/L, cyan—80 mg/L. Diffuse reflectance spectra of GO, TCPP-GO and THPP-GO supramolecular assemblies isolated as a solid phase (**c**).

**Figure 3 molecules-24-00688-f003:**
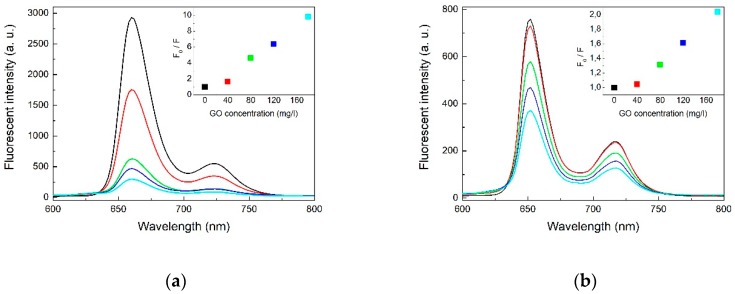
Fluorescence spectra of two metal-free porphyrins THPP (**a**) and TCPP (**b**), with different concentrations of GO red—20 mg/L, green—40 mg/L, blue—60 mg/L, cyan—80 mg/L.

**Figure 4 molecules-24-00688-f004:**
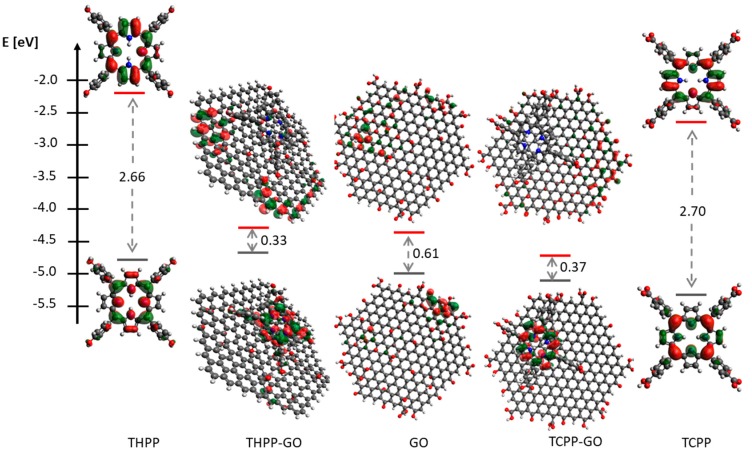
The energies of HOMO and LUMO orbitals of graphene oxide, studied porphyrins and their supramolecular assembly. Geometry of all these structures has been optimized at b3lyp/3–21 g level of theory.

**Figure 5 molecules-24-00688-f005:**
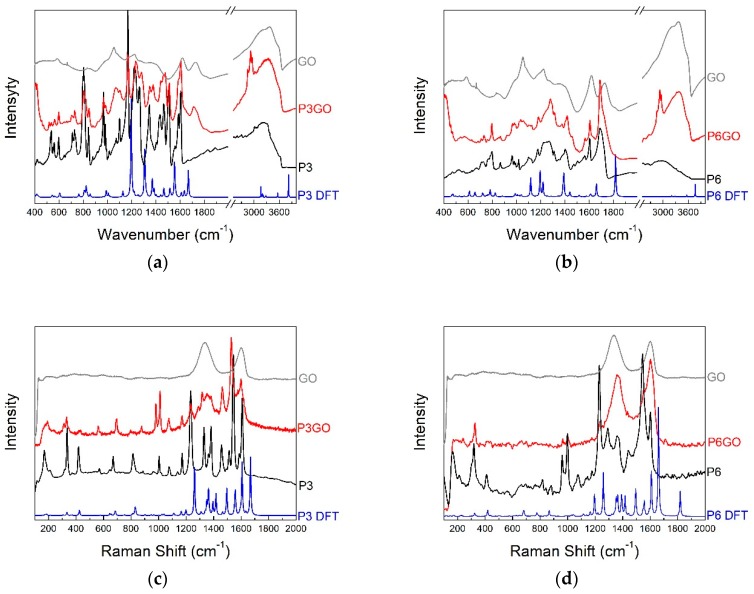
The calculated (blue) and experimental IR absorption (**a**,**b**) and Raman scattering (**c**,**d**) spectra of two metal-free porphyrins THPP and TCPP (black) and the experimental spectra of GO (gray) and hybrids THPP-GO and TCPP-GO (red) spectra at room temperature.

**Figure 6 molecules-24-00688-f006:**
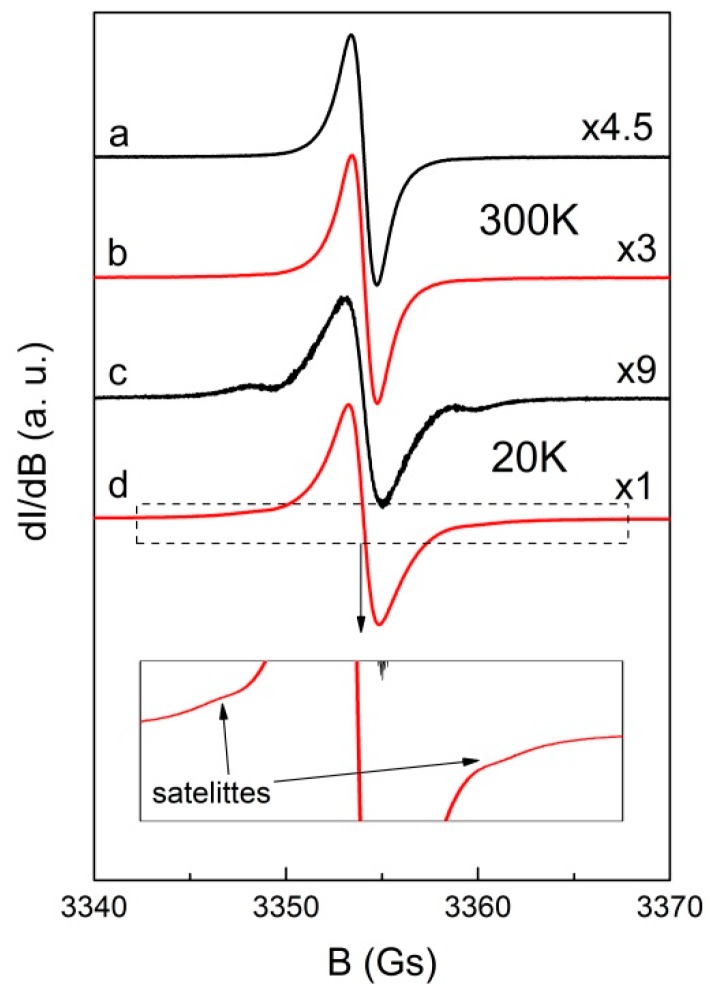
Electron paramagnetic resonance (EPR) spectra of powdered GO recorded at 300 and 20 K obtained after evaporation of sonificated GO from H_2_O (**a**,**c**) and GO from H_2_O&THF (**b**,**d**) solution. The numbers on the right side denote the relative amplifier gain and insert indicates the satellite lines.

**Figure 7 molecules-24-00688-f007:**
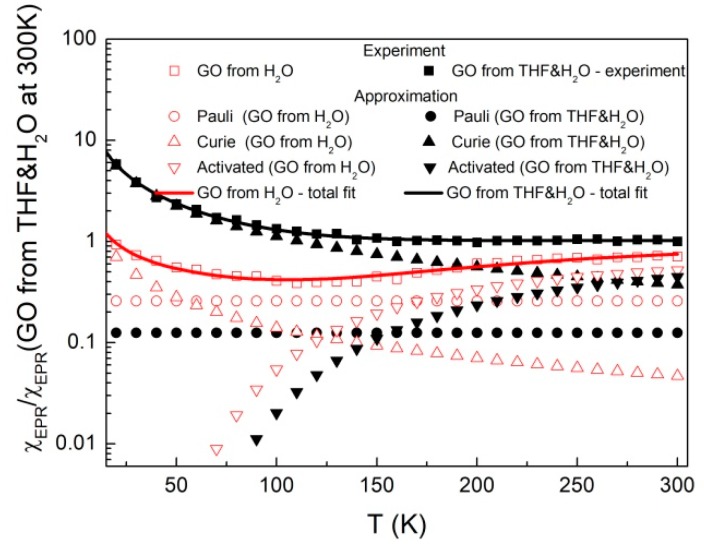
Spin susceptibility of powdered GO after evaporation of sonificated GO from H_2_O and H_2_O&THF solution.

**Figure 8 molecules-24-00688-f008:**
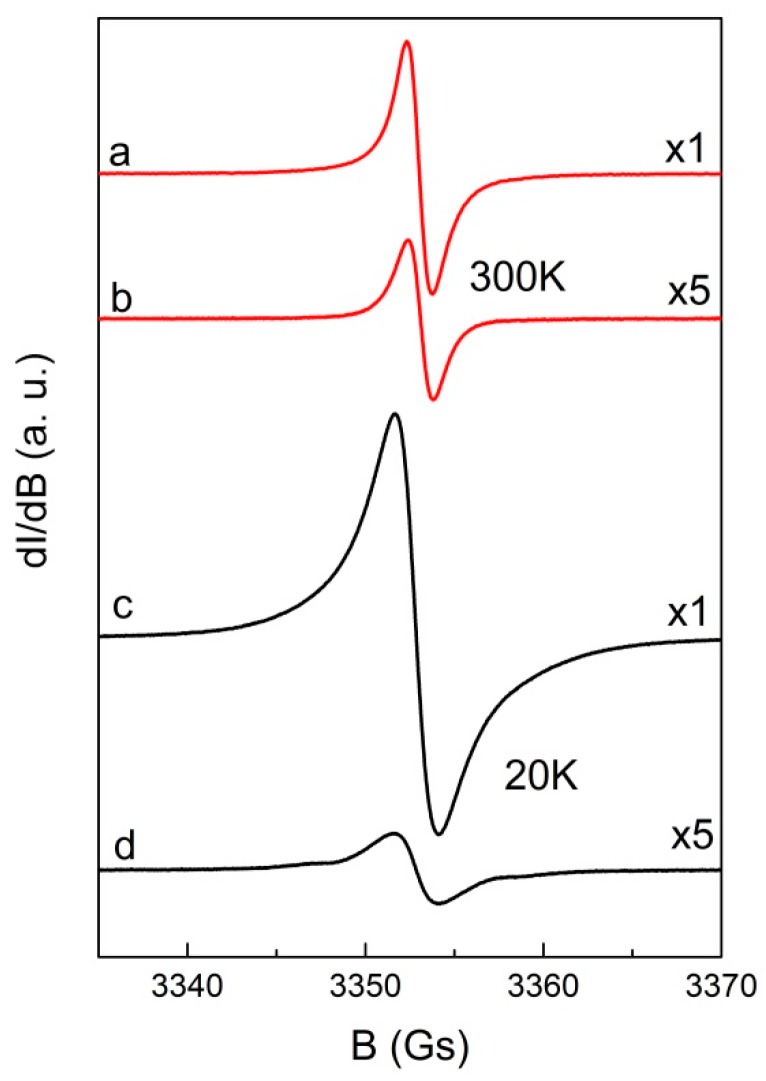
EPR spectra of THPP-GO from H_2_O&THF (**a**,**c**) and TCPP-GO from H_2_O & THF (**b**,**d**) recorded at 20 and 300 K. The numbers on the right side denote the relative amplifier gain.

**Figure 9 molecules-24-00688-f009:**
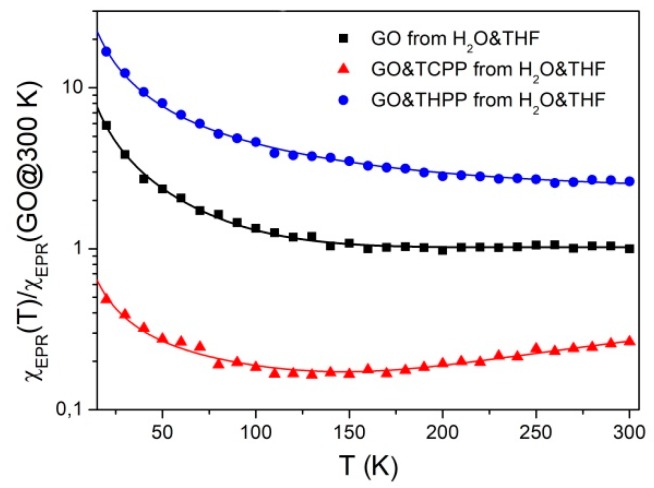
Comparison of the relative spin susceptibility of GO, porphyrin and hybrid system of THPP-GO and TCPP-GO. Symbols indicate experimental points and lines are fits according to Equation (2).

**Figure 10 molecules-24-00688-f010:**
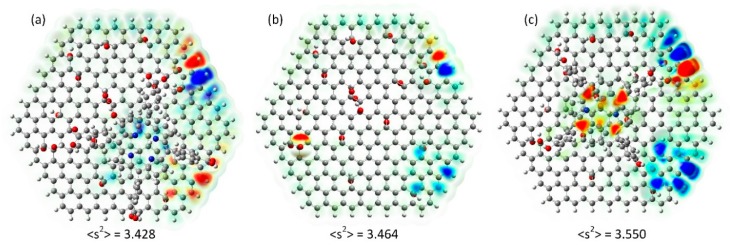
Computed spin density projected in the charge density isosurface for the TCPP-GO complex (**a**), pristine graphene oxide (**b**) and the THPP-GO complex (**c**). Number on figures indicate calculated values for the sum of squared spin densities of individual atoms. Red areas correspond to spin α and blue ones to spin β.

**Figure 11 molecules-24-00688-f011:**
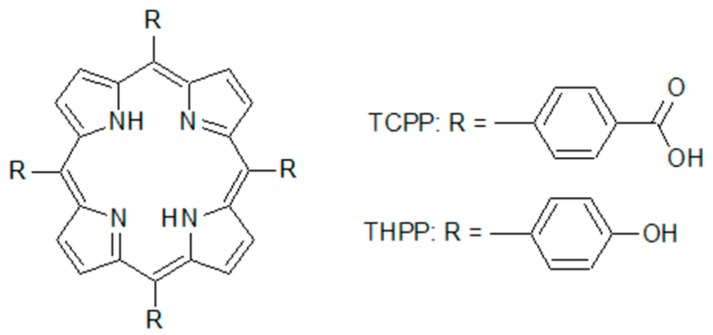
The molecular structures of THPP and TCPP.

**Table 1 molecules-24-00688-t001:** Selected characteristic vibrational features of THPP and THPP-GO: s—stretching, b—bending, w—wagging, r—rocking, def.—deformation.

IR (cm^−1^)	Raman (cm^−1^)	DFT (cm^−1^)	Bands Assignment
THPP/THPP-GO	THPP/THPP-GO	THPP	
	334/330	333	breathing porphyrin ring
	417	424	def. benzene ring
535/536		542	C-H w in benzene ring + def. porphyrin
560/566		568	C-H w in benzene ring + def. porphyrin
597/599		607	breathing benzene ring + def. porphyrin
	647	645	N-H w
	669/695	687	N-H w + C-H w
729/729		764	N-H w
804/805		824	C-H w + N-H w
	814	835	C-C-C b in benzene ring + N-H w + C-O s
843/846		857	C-H w
966/968		989	C-C s in porphyrin
983/985		1008	C-C s in porphyrin
	1004/1010	1027	C-C s in porphyrin
	1018	1039	C-H r + N-H r
1075/1068	1075/1075	1109	C-H r
1100/1102		1128	C-H r
	1139	1165	N-H r + C-H r
1169/1170		1197	C-O-H b + C-H r
	1171/1171	1199	C-O-C b + C-H r
1223/1233		1221	N-H r + C-H r + C-O-H b
	1234/1234	1261	C-C s between porphyrin and aryl group
1263/1281		1308	C-O s + C-H r
1346/1350		1370	C-H r + C-O-H b
	1363	1395	C-C s
	1381/1383	1417	C-N s + C-H r
1402/1442		1439	C-C s + C-H r
1433/1463		1466	C-H r + C-O-H b
	1459/1452	1496	C-C s + C=C s
1465/1477		1515	C=C s
1508/1511		1555	C-C s + C-H r + C-C s between porphyrin and aryl group
	1516/1527	1559	C-C s + C=C s + C-C s between porphyrin and aryl group
	1544	1607	C=C s + C-N-H b
	1558	1610	C=C s + C-N-H b
1586/1588		1635	C-C s + C-O-H b
1605/1606	1608/1601	1668	C-C s + C=C s
1662/1648			C=N s
-/1731			C=O s in GO

**Table 2 molecules-24-00688-t002:** Selected characteristic vibrational features of TCPP and TCPP-GO: s—stretching, b—bending, w—wagging, r—rocking, def.—deformation.

IR (cm^−1^)	Raman (cm^−1^)	DFT (cm^−1^)	Bands Assignment
TCPP/TCPP-GO	TCPP/TCPP-GO	TCPP	
	319/324	323	breathing porphyrin ring
	410	418	def. benzene ring
	671/690	687	N-H w
723/732		763	N-H w
797/796		824	N-H w+ C-H w
	817	837	C- C-C b + N-H w
866/866		884	C-H w + def. pophyrin
964/968		989	breathing porphyrin and benzene rings
980/982		1008	breathing porphyrin ring
994/994		1022	C-N s + C-H r + N-H r
	999/1003	1027	C-C s
1019/1023		1039	C-H + N-H r
	1073	1114	C-H r
1101/1101		1118	C-O s + C-H r
	1145	1164	N-H r
1176/1181	1176	1195	C-H r + C-O-H b
1221/1225		1220	C-H r + C-O-H b + N-H r + C-N s
1226	1231/1236	1258	C-C s between porphyrin and aryl group
1270/1279		1287	C-C s + C-N s + N-H b
	1293	1363	C-H r
1310/1314	1318	1391	C-O-H b + C-C s + C-N s + C-H r
	1358/1363	1417	C-C s + C-N s + C-H r
1403/1418		1441	C-C s + C=C s + C-H r
	1440	1495	C-C s + C-H r i N-H r
1473/1470		1518	C=C s w
	1495	1558	C=C s
1505		1579	C=C s + C-C s + C-H r + N-H r
	1545/1555	1609	C=C s
1564/1567		1614	C=C s
1605/1607	1605/1603	1660	C=C s + C-C s
1691/1690		1661	C=O s
1727/1720		1818	C=O s in GO

**Table 3 molecules-24-00688-t003:** Parameters of spin susceptibility obtained from approximation of Equation (2) to experimental points.

Sample	*C* _1_	*C* _2_	*ΔE*/*k*_B_ [*K*]	*A*	Total Spin Concentration at RT [10^17^/g]	Concentration of Delocalized Electrons [10^17^/g]
GO from H_2_O	14.1 ± 0.8	1088 ± 168	629 ± 47	0.26 ± 0.02	2.33	0.89 ± 0.07
GO from H_2_O&THF	112.6 ± 1.8	838 ± 156	504 ± 58	0.13 ± 0.05	3.44	0.45 ± 0.17
THPP-GO from H_2_O&THF	318 ± 4	1111 ± 822	946 ± 355	1.33 ± 0.07	9.02	4.58 ± 0.24
TCPP-GO from H_2_O&THF	7.96 ± 0.28	945 ± 326	951 ± 101	0.108 ± 0.006	0.91	0.37 ± 0.02

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
