# Peer review of "Supramolecular Complexes of Graphene Oxide with Porphyrins: An Interplay between Electronic and Magnetic Properties"

_molecules, 2019, doi:10.3390/molecules24040688_

Round 1

Reviewer 1 Report

The manuscript by Szacilowski et al. describes the formation of supramolecular complexes between graphene oxide and porphyrins. The authors analyzed the process by using several spectroscopies. I understand there are some spectroscopic change accompanying the complex formation. However I am conscious of following points.

1)    The authors estimated the frontier orbitals of GO and two TPP derivatives by DFT calculation using 3-21 level basis set. I think this level is not enough to characterize the electronic structure of these large pi-systems.

2)    Please add the solvent for optical spectroscopies. Based on the computational study, authors consider the driving force for the supramolecular complexation. As both GO and two porphyrin derivatives have H-bonding sites (-COOH and -OH), there is an alternative possibility of aggregation induced by H-bonding. I guess the comparison of solvents is important for these kind of research in connection with fluorescent property.

3)    In this manuscript the author dissolved materials by sonification in water or water & THF. Sonification often gave a spin center by bond scission or formation of hydro radical from water (for example DOI: 10.1016/0032-3861(92)90516-Y ). Also GO has intrinsically spin site. I appreciate the addition of these data or description to elucidate the change in the magnetic characteristics during the complexations.

Reviewer 2 Report

In this paper, supramolecular complexes of graphene oxide with 2 porphyrins: an interplay between electronic and 3 magnetic properties of the authors Kornelia Lewandowska et. al., report an interesting study of the Graphene oxide (GO) modified by THPP and TCPP: As well as a supramolecular, optical, magnetic properties and DFT calculations of the of the GO, and GO-porphyrin molecules. In my opinion this manuscript is interesting. Authors using different techniques (IR, Raman, EPR and DFT calculations) to explain the structural, electronic and topological properties of THPP and TCPP with GO.

Therefore, I recommend publication of this work after some revision as follows:

1.     What is the meaning of word: complexes in the title, abstract, as well as in the whole manuscript, because the investigation is about interaction between GO and two porphyrins, which are free of metals, so there is not complexes formation. I suggest the authors remove the phrase "metal-free", because is obvious the author report the chemistry modification of the GO with porphyrins and did not form any complexes with metal ions

2.     The authors must clarify maybe by one scheme, which interactions or which groups of GO are modified or interact with porphyrins; is a non-covalent interaction, covalent bond, etc. because the vibration analysis (line 268 the write the following: “These changes may stem from stiffening porphyrin molecule located above the plane of the graphene oxide”, where come from this statement. The characterization by spectroscopy gave them information about the structure conformation? please explain this.

3.     In the abstract the authors statement “The FTIR and Raman scattering measurements confirm nanoflakes and porphyrin molecules. What is the evidence from IR and Raman, that allow conclude that the porphyrins adopt a conformation p-stacking between hydrophobic regions of GO,” because this is an interaction non-covalent. This is important since the title is about supramolecular.

4.     Line 67 the paragraph is repeat the paragraph: “The noncovalent stacking of aromatic organic molecules on graphene oxide through π-π interaction is emerging as a promising route to tailor the electronic properties of graphene oxide”

5.     The figure 5 what is the meaning of P3, P3GO, P6GO, etc.

6.     They need to refer the optical absorption measurements in powder to determine the presence of porphyrins composites with GO and not only in solution. (Reference Zare-Dorabel RSC adv. 2015)
